# The Inducible Role of Ambient Particulate Matter in Cancer Progression via Oxidative Stress-Mediated Reactive Oxygen Species Pathways: A Recent Perception

**DOI:** 10.3390/cancers12092505

**Published:** 2020-09-03

**Authors:** Chiang-Wen Lee, Thi Thuy Tien Vo, Ching-Zong Wu, Miao-Ching Chi, Chieh-Mo Lin, Mei-Ling Fang, I-Ta Lee

**Affiliations:** 1Department of Orthopaedic Surgery, Chang Gung Memorial Hospital, Puzi City, Chiayi County 613, Taiwan; cwlee@mail.cgust.edu.tw; 2Department of Nursing, Division of Basic Medical Sciences, Chronic Diseases and Health Promotion Research Center, Chang Gung University of Science and Technology, Puzi City, Chiayi County 613, Taiwan; 3Department of Safety Health and Environmental Engineering, Ming Chi University of Technology, New Taipei City 243, Taiwan; 4College of Medicine, Chang Gung University, Guishan District, Taoyuan City 333, Taiwan; 5School of Dentistry, College of Oral Medicine, Taipei Medical University, Taipei 110, Taiwan; d204108005@tmu.edu.tw (T.T.T.V.); chinaowu@tmu.edu.tw (C.-Z.W.); 6Chronic Disease and Health Promotion Research Center, Chang Gung University of Science and Technology, Chiayi County 613, Taiwan; mcchi@mail.cgust.edu.tw; 7Division of Pulmonary and Critical Care Medicine, Chiayi Chang Gung Memorial Hospital, Chiayi County 613, Taiwan; f124510714@cgmh.org.tw; 8Department of Respiratory Care, Chang Gung University of Science and Technology, Chiayi County 613, Taiwan; 9Department of Nursing, Chang Gung University of Science and Technology, Puzi City 613, Taiwan; 10Graduate Institute of Clinical Medical Sciences, College of Medicine, Chang Gung University, Taoyuan 333, Taiwan; 11Center for Environmental Toxin and Emerging-Contaminant Research, Cheng Shiu University, Kaohsiung 833, Taiwan; k6764@gcloud.csu.edu.tw; 12Super Micro Research and Technology Center, Cheng Shiu University, Kaohsiung 833, Taiwan

**Keywords:** cancer, particulate matter, oxidative stress, antioxidant, redox signaling pathways, carbon monoxide releasing molecules

## Abstract

**Simple Summary:**

Particulate matter, especially the fine fraction PM2.5, is officially stated as carcinogenic to human. There are compelling evidences on the association between PM2.5 exposure and lung cancer, and there are also some preliminary data reporting the significant links between this fraction with non-lung cancers. The underlying mechanisms remain unclear. Further studies related to such scope are highly required. The purpose of this work is to systemically analyze recent findings concerning the relationship between PM2.5 and cancer, and to thoroughly present the oxidative stress pathways mediated by reactive oxygen species as the key mechanism for carcinogenesis induced by PM2.5. This will provide a more comprehensive and updated knowledge regarding carcinogenic capacity of PM2.5 to both clinicians and public health workers, contributing to preventive and therapeutic strategies to fight against cancer in human.

**Abstract:**

Cancer is one of the leading causes of premature death and overall death in the world. On the other hand, fine particulate matter, which is less than 2.5 microns in aerodynamic diameter, is a global health problem due to its small diameter but high toxicity. Accumulating evidence has demonstrated the positive associations between this pollutant with both lung and non-lung cancer processes. However, the underlying mechanisms are yet to be elucidated. The present review summarizes and analyzes the most recent findings on the relationship between fine particulate matter and various types of cancer along with the oxidative stress mechanisms as its possible carcinogenic mechanisms. Also, promising antioxidant therapies against cancer induced by this poison factor are discussed.

## 1. Introduction

Cancer is a generic term given to a set of malignant diseases or illnesses in which cells located in any body sites become abnormal and keep multiplying without control, producing tumors. As these tumors grow up, an invasion process in which cancer cells spread into adjacent tissues, or even metastasis in which cancer cells migrate to distant locations in the body via either bloodstream or lymphatic system to generate new tumors, might occur. Due to its uncontrollable and invasive characteristics, cancer is well-established as being severely life-threatening. Thus, despite the diversity in incidence and mortality, cancer is one of the leading causes of both premature death and overall death all over the world, adversely affecting individual lifespans as well as global economics. In fact, carcinogens, or substances that generate cancer, exist around us in various forms, including chemicals, physical elements, and oncogenic organisms. On the other side, air pollution, which is considered as the contamination of the indoor and outdoor air by a range of gases and solids, is a severe worldwide problem due to its variety of deleterious impacts, especially on human health. Recently, outdoor air pollution, along with its major component which is so-called particulate matter, have been classified as carcinogenic to humans by the International Agency for Research on Cancer (IARC), a specialized cancer agency of the World Health Organization [1]. In fact, everyone breaths thousands of liters of air daily. This implies that humans might be at greater risk of developing cancer if continuously exposed to significant concentrations of polluted air, particularly particulate matter, over time. Particulate matter, or PM, is a complex mixture of varying airborne solid and liquid particles of organic and inorganic substances in the ambient atmosphere [2]. Over the decades, there have been numerous pieces of evidence indicating the positive relationship between diverse degrees of health compromises with both short-term and long-term PM exposure. Several studies found that the size of PM, as well as its surface area, probably determine the biological effects, such as inflammatory responses, oxidative damage, and other phenomena [3,4,5,6,7]. Based on the aerodynamic diameter of the particles, PM is classified into coarse PM (PM10), fine PM (PM2.5), and ultrafine PM (PM0.1) that is no greater than 10 μm, 2.5 μm, and 0.1 μm in diameter, respectively [8]. Generally, PM is introduced into the human body through the inhalation process. As it enters into our body, coarse PM is retained in nasal cavities and upper airways, whereas fine and ultrafine PM might penetrate deeper into the lungs and bronchi alveoli, and even further travel to distant organs or elicit systemic effects through the circulation [9] (Figure 1). In other words, the smaller the particles are, the worse impact they might induce. Indeed, PM0.1, which is the finest fraction of particles, appears to pose the most threatening risks to human health, as it not only easily penetrates into lungs and subsequently translocates to other vital organs, but also remains longer in body parts because of its extremely small aerodynamic diameter. Moreover, this ultrafine fraction possibly causes more severe conditions due to its greater surface area and higher concentrations of absorbed noxious constituents. The associations between exposure to PM0.1 and various illnesses related to the respiratory system, cardiovascular system, central nervous system, or even cancer have been reported [10]. Still, the literature on the adverse health effects caused by PM0.1 remains scarce compared to PM10 and PM2.5. This paucity of data might be attributed to the lack of standardized methods and protocols in estimating the outcomes upon exposure to PM0.1. Thus, although the potential of PM0.1 to cause damage to human health is undoubtful, its precise role involving in the pathogenesis of many diseases is yet to be elucidated. It is essential to carry out more investigations in the future to clarify this scope. Currently, PM2.5 is chosen to provide international recommendations regarding health effects because it might best describe the components of PM responsible for adverse conditions [11]. Furthermore, the deleterious effects of different PM fractions are overlapping due to the overlying in particle sizes among fractions. Otherwise speaking, PM2.5 might partly elicit similar effects to those of smaller PM0.1. Therefore, PM2.5 has still been the major particle fraction that is of research in relation to the direct consequences of inhaled PM on human health. Studies related to the health effects of PM2.5 have been increased widely and deeply, providing evidence about its potential in the pathogenesis of lung cancer, as well as several types of non-lung cancer [12,13,14]. However, the underlying mechanisms remain unclear. To our knowledge, there is no specific review regarding the impact of PM2.5 as a single pollutant on general cancer progress. As a result, the present review aims to summarize and analyze the most recent findings in this area; specifically, the associations between PM2.5 and various types of cancer along with the oxidative stress mechanisms as its possible carcinogenic mechanisms. From the viewpoint of oxidative stress mechanisms, promising antioxidant therapies against PM2.5-related cancers are also discussed.

## 2. Associations between Fine Particulate Matter and Cancer Progression

### 2.1. Epidemiological Evidences of PM2.5-Associated Lung Cancer

According to the latest global cancer data published by IARC, lung cancer is the top-ranking cause of cancer incidence and mortality worldwide. It is responsible for the highest number of either new cases or deaths, approximately 11.6% and 18.4% of the total, respectively [15]. So far, while smoking or consumption of tobacco products is the main cause of lung cancer that accounts for about 90% of cases; other risk factors, including environmental and occupational exposure, might contribute to the etiology of lung cancer [16,17]. As mentioned above, PM2.5 easily penetrates deeper into the lungs or bronchi alveoli and persistently remains in these areas due to its extremely small size. Thus, the most vulnerable organ that is compromised by exposure to these invisible particles is the lung. On the other hand, the conclusion that the increased risk of lung cancer associated with the increasing levels of exposure to PM was officially claimed by IARC [1]. Since then, numerous epidemiological studies have further demonstrated the positive relationship between PM2.5 and lung cancer. Measurements of disease (incidence) and/or death (mortality) within populations are important to evaluate the association between risk factors with a specific disease [18]. However, for lung cancer, it is well-established that the survival rate is relatively low and probably drops from 59% in stage 1 to only 5.9% in stage 4 [19]. Due to this high case-fatality rate, mortality might be considered as comparable as incidence for lung cancer. As a result, the majority of studies have investigated the influence of PM2.5 on lung cancer mortality. Based on 17 cohort studies (from 2008 to 2013) and one case-control study of lung cancer, including four studies from Europe, eight studies from North America, and two studies from other regions, Hamra and coworkers performed the overall meta-estimates for PM2.5 [20]. The authors found that lung cancer mortality increases by 9% for every 10 μg/m^3^ PM2.5 increase, consistent with a subsequent meta-analysis which included 19 cohort studies [21]. However, there are two main limitations of these publications. Firstly, they collected mortality and incidence studies together, which might cause inaccurate calculation and interpretation. Secondly, almost all of studies were carried out in developed countries where the annual medium concentration of PM2.5 is greatly lower than that in developing countries, thus inappropriately representing the global scenario. To fill these gaps, one meta-analysis of 30 cohort studies between 1999 and 2017 that involved more than 1 million cases in 14 countries all over the world was conducted [22]. The key strength of this study is the separation of cancer mortality from cancer incidence. The results showed that long-term exposure to PM2.5 also increases mortality from lung cancer but at higher risk, 14% per 10 μg/m^3^ increase in PM2.5. Collectively, it could be concluded that PM2.5 is prospectively related to a significantly increased risk of lung cancer mortality. Actually, not every patient who suffers lung cancer would die by this disease; it is thus important to separately determine the impact of PM2.5 on the development of lung cancer. Since urbanization coupled with industrialization originally progressed in North America and Western countries, the earlier reports on PM2.5-associated lung cancer incidence came from these population groups. The positive relationship between long-term PM2.5 exposure and lung cancer incidence was demonstrated in three large-scale prospective cohort studies. The first report was the Canadian population-based case-control study, which included 2390 incident cases of histologically diagnosed lung cancer in 8 out of 10 provinces between 1994 and 1997, in order to estimate the ambient air pollution (fine PM, nitrogen dioxide, and ozone)-related lung cancer incidence over 20-year-exposure [23]. The second report was the United States population-based cohort using Nurse’ Health Study (NHS) data to investigate the lung cancer incidence from 1994 through 2010 in relation with 72-month average exposures to outdoor PM, but specifically focused on females [24]. The third report was the European population-based meta-analysis, which collected 17 cohort studies in 9 European countries during a 12.8-year average follow-up from The European Study of Cohorts for Air Pollution Effects (ESCAPE) data in order to determine the association between long-term exposure to ambient air pollution and lung cancer incidence [25]. After full adjustment, all three analyses consistently confirmed the positive relationship between long-term exposure to PM2.5 and incidence for lung cancer. The former work showed the increase in lung cancer incidence was 1.29 (OR = 1.29, 95% CI = 0.95–1.76) with 10 μg/m^3^ increase in PM2.5, whereas the two latter studies estimated the hazard ratio of 1.06 (HR = 1.06, 95% CI = 0.91–1.25) and 1.40 (HR = 1.40, 95% CI = 0.92–2.13) per 10 μg/m^3^ of PM2.5, respectively. In the present, Asian countries, especially China, have become hot spots in terms of air pollution in the world. Thus, the majority of recent studies on PM2.5-associated lung cancer incidence have been carried out in these areas. A meta-analysis by Huang et al. (2017) [26] found that the lung cancer incidence associated with PM2.5 was greatest in Asia, followed by North America, and then Europe. This might be due to the greater population densities and higher levels of PM2.5 derived from the urbanization combined with industrialization in Asia compared to that in North America and Europe. The latest report from China is a nationwide study in 295 Chinese counties about the effects of ambient PM2.5 exposure on annual incidence rates of lung cancer for both genders in China. In this study, the mean incidence rates of lung cancer for males and females were 50.38 and 22.16 per 100,000 people, respectively, whereas the annual mean PM2.5 value was 43.02 μg/m^3^. Then, it was demonstrated that for an increment of 10 μg/m^3^ PM2.5, the incidence rates of lung cancer change by 3.57% and 2.71% compared to the mean values for males and females, respectively [27]. Previously, a prospective analysis of 89,234 Canadian women who participated between 1980 and 1985 to determine the incidence of lung cancer through 2004 was conducted [28]. Although the PM2.5 concentration was relatively low with a mean value of 9.50 μg/m^3^, the increment of 10 µg/m^3^ PM2.5 was associated with an increased risk of lung cancer of 34%, which is considerably stronger than the mentioned estimation from China. Guo and colleagues suggested two possible explanations for this conflict. It firstly might be due to the high proportion of less toxic crustal materials and dust in PM2.5 compositions in China. Furthermore, the proportion of elderly, who are more vulnerable to PM2.5 exposure, in China is still significantly lower than that in other developed countries [27]. Taken together, existing findings have demonstrated a positive association between PM2.5 and the incidence of lung cancer, even at low-level concentrations. In addition to mortality and incidence rates, it is important to examine the lag time between the previous exposure to risk factors and definite diagnosis since the cancer process might have been asymptomatically triggered for a long time. However, the lag effects of PM2.5 on the risk of lung cancer have been largely overlooked. Limited literature suggested the long-term lag effects of PM2.5 on lung cancer incidence, subsequently highlighting the possibility that the incidence of lung cancer is correlated with both current and previous exposure to PM2.5 [27].

In summary, while epidemiology has produced a variety of strong evidences on PM2.5-associated lung cancer, there are three major gaps that require further study to provide an in-depth understanding on the influence of these invisible particles. Firstly, there are diverse air pollutants existing in the ambient atmosphere that might synergistically affect the human body. This might lead to the overestimation of the actual effects of PM2.5. Secondly, there are relatively few cohort studies that examine the long-term effects of PM2.5 on lung cancer incidence because of the restrictions in available data. Simultaneously, cross-sectional studies that show PM2.5 effects that might be ignored in cohort studies are also quite sparse. Finally, exclusive evaluation of current PM2.5 exposure might underestimate the actual impact of PM2.5 due to the lack of lag effect observations.

### 2.2. Experimental Evidences of PM2.5-Associated Lung Cancer

Estimation of the amounts of deposited particles in the human respiratory tract upon PM2.5 inhalation is a crucial prerequisite for the assessment of the hazardous risks associated with exposure to this pollutant. Although the mechanisms and dynamics of particle deposition and clearance have been widely studied, the deposition and clearance as well as the invasion to other tissues of single PM2.5 have been yet to be clearly examined. Based on the dosimetric model of the human respiratory tract, the so-called ICPR Publication 66, which was issued by The International Commission on Radiological Protection (ICRP) in 1994 [29], researchers have been able to calculate the pulmonary deposition of PM10 and PM2.5, contributing to clarify the deposition patterns of inhaled particles in the airway [30,31]. However, this model ignored the kinematic difference between particles with diverse sizes. It means different particles were assumed to deposit uniformly in every region of the respiratory tract. Furthermore, theoretical models could never completely replace the in vivo data on the pulmonary deposition of particles since the actual deposition is determined by a variety of factors, such as concentrations and characteristics of particles, respiratory physiology, lung morphology, and flow parameters [32,33]. Thus, real-time observations with high temporal and spatial resolutions ought to be addressed in in vivo experiments. Recently, Li et al. reported an in vivo measurement using a fluorescent imaging method without animal sacrifice to visualize the deposition process of inhaled particles with high resolution in terms of time and location [34]. This work found that the deposition pattern of PM2.5 is variable and the maximum deposition rate of the particles is significantly higher than the estimation from the dosimetric model used in previous studies. Despite the inconsistent results, both theoretical and experimental studies indicated the capacity of PM2.5 to remain in lungs, or even be absorbed into the bloodstream or be transported to extrapulmonary organs, in turn exerting its local toxicity or systemic effects, or both. It is well-established that cell proliferation, migration, and invasion are crucial steps of cancer progression. Using adenocarcinomic human alveolar basal cell line A549 and non-small lung carcinoma cell line H1299, Yang et al. provided in vitro evidence on the possibility of PM2.5 to exert non-small-cell lung cancer [35]. This work suggested the potentials of PM2.5 exposure in promoting viability, proliferation, migration, and invasion of tumor cells. Similarly, Wei et al. found that in PM2.5 exposure, both acute and chronic models might enhance in vitro migration and invasion of A549 cells as well [36]. In 2018, Xu and coworkers carried out another study which also used the A549 cell line to investigate the effects of PM2.5 exposure on lung cancer cells [37]. It revealed that PM2.5-treated tumor cells produce exosomes that enhance the cell proliferation, but have no impact on cell migration and cell invasion in vitro. In addition to proliferative and invasive characteristics of PM2.5-exposed cancer cells, epithelial mesenchymal transition (EMT) is an important event during tumor development. Exposure to PM2.5 could induce in vitro EMT in A549 and H292 (human mucoepidermoid pulmonary carcinoma) cells [36,38]. In animal experiments, Yang et al. fed two groups of female Kunming mice in either a laboratory or underground parking lot as a control group and polluted group for 3 months [39]. For the treated group, the lung tissues were damaged, and the amounts of inflammatory cells as well as the levels of inflammatory cytokines were increased. Moreover, the numbers of macrophages were considerably greater in mice lung tissues after treatment. These results suggested that PM2.5 might trigger the inflammatory responses and cause an immune disturbance, resulting in lung injury in mice. Additional studies also found that exposure to PM2.5 promoted lung cancer development in tumor-bearing mice [40].

Together, in vitro and in vivo experimental evidence further proves the strong association between PM2.5 and lung cancer. In particular, these particles might exert adverse effects on promoting lung cancer progression, as well as inhibiting immune defenses. However, the underlying mechanisms are yet to be clearly understood.

### 2.3. Literature Evidence of PM2.5-Associated Non-Lung Cancer

It is well-believed that every substance optimally exerts its biological effects in one organ if it targets and remains in that body part. For PM2.5, it can enter the lower airway to reach the lungs and even the alveolar areas due to its small aerodynamic diameter. Moreover, finer particles or aerosols, which carry toxic substances, might be absorbed into the bloodstream or delivered to extrapulmonary organs through blood circulation. Experimentally, upon in vivo inhalation work, single particles were observed in mice kidney and liver using the fluorescent imaging method [34]. Therefore, while exposure to PM2.5 has been extensively demonstrated to be associated with lung cancer, it is likely to be a risk factor for other types of cancer. There have been growing but inconsistent findings related to this suspicion. So far, a few studies have provided evidence on the adverse effects of PM2.5 exposure towards cancer progress in almost of all vital organs aside from the lungs in the human body, including the digestive tract and accessory organs such as oral and oropharyngeal organs [41,42,43], esophageal organs [42,43], the stomach [42,44,45], colorectal organs [43,45,46,47], the liver [43,45,47,48,49,50], and the pancreas [43,49], female-specific organs such as breasts [43,45,51,52], cervix [45], and genital organs [43], urinary organs such as kidneys [46] and the bladder [45,46], lymphoid [45] such as Hodgkin and non-Hodgkin lymphoma as well as leukemia [53], and others such as laryngeal organs [42]. Here, the current review made up a collection of existing data about the statistically significant associations between the exposure to increased concentrations of PM2.5 with the risk of various types of non-lung cancer, which was illustrated in Figure 2.

While available data have shown the possibility of PM2.5 as a risk factor of cancer mortality and/or incidence for a variety of extrapulmonary organs, a few studies have shown the positive but not significant relationship or even no correlation between PM2.5 exposure and non-lung cancer [54,55,56,57,58]. This might be due to the occurrence of limitations or gaps making the existing evidence weak and inconsistent. The main gap is the wide discrepancies in the scope of study and the geographical areas among reports. Another gap is the undesirable contribution of confounders due to improper control of the bias. Besides, the lack of documentation from experimental and mechanism studies is a significant gap as well. Therefore, cross-sectional combined with prospective cohort studies at the large-scale, such as nationwide or multi-center studies, as well as experimental evidence are essentially required to further investigate the actual effects of PM2.5 on the risk of different types of non-lung cancer.

## 3. Association between Fine Particulate Matter Induced-Oxidative Stress Mechanisms and Lung Cancer Process 

Whilst the associations between PM2.5 exposure and extrapulmonary cancers are yet to be strongly confirmed, there has been solid evidence on the relationship linking this pollutant with lung cancer processes. However, the underlying mechanisms for this connection still remain unclear and inconclusive. One of the possible mechanisms which has been widely accepted by a variety of extensive studies is the oxidative stress paradigm through the production of reactive oxygen species as a result of long-term exposure to fine PM. Thus, this part would cover the available literature on the oxidative stress mediated by PM2.5-induced ROS which is involved in lung carcinogenesis. The concept of oxidative stress was first introduced in the field of oxygen-related damage in biological systems by Helmut Sies in 1985 [59]. From this point of view, it can be best defined as the following description by the same scholar: “Oxidative stress is an imbalance between oxidants and antioxidants in favor of the oxidants, leading to a disruption of redox signaling and control and/or molecular damage” [60]. This stress might impair vital cellular structures, such as membrane lipids, carbohydrates, proteins, enzymes, DNA and RNA, contributing to the pathogenesis. On the other hand, reactive oxygen species or ROS, which include both free radicals such as superoxide anion and hydroxyl radical as well as non-radicals such as hydrogen peroxide, are the most greatly generated and possess high oxidative capacity. Physiologically, ROS are normal by-products of cellular processes, and a cellular defense that contains free radical scavenger and antioxidant enzymes is able to function against ROS to maintain the redox equilibrium in the cells. An excessive or sustained increase in ROS generation, which is improperly neutralized by antioxidant protection due to deficiency or exhaustion, would result in oxidative stress and subsequent pathological injury [61]. Ambient fine PM has been demonstrated as a complicated and heterogeneous mixture of inorganic and organic constituents, in which heavy and transition metals with redox properties, redox-cycling quinones, polycyclic aromatic hydrocarbons (PAHs), volatile organic compounds (VOCs), and other chemicals all can synergistically enhance the production of intracellular ROS. In addition to the direct generation induced by PM2.5, ROS might be also released by host cells such as macrophages following the interaction with this pollutant. For example, macrophages in the airways and the alveolar areas, which significantly play protective roles in the respiratory tract, possibly generate ROS upon phagocytosis of inhaled particles [61]. Thus, in the case of lungs, the production of ROS is likely to be dramatically increased by chronic exposure to fine PM, leading to oxidative stress. This stress might be involved in the activation of signaling pathways, the modulation of transcription factors, and the impairment of DNA, which are not only the hallmarks of cancer initiation but also regulate essential events of cancer development such as proliferation, apoptosis, angiogenesis, and metastasis (Figure 3). The adverse biological effects involved in lung carcinogenesis caused by PM2.5 exposure are shortly presented in Table 1 as well.

### 3.1. Oxidative Stress-Activated Signaling Pathways

There has been growing evidence that PM2.5-induced ROS production capably triggers a number of redox-sensitive signaling pathways that result in diverse biological processes in human lungs. The mitogen-activated protein kinases (MAPKs) pathway is an important intracellular signal-transduction system involved in response to extracellular stimuli. There are three main members of MAPKs family, including ERK1/2, JNK, and p38. Besides, phosphatidylinositol 3-kinase/protein kinase B (PI3K/AKT) pathways might function as a regulatory paradigm of MAPK activity. Together, these protein kinase pathways are crucial for the regulation of cell survival and proliferation, as well as inflammatory or other reactions in response to oxidative stress [77]. It was demonstrated that PM2.5-induced oxidative stress-activated JNK, ERK1/2, p38 and AKT pathways in human epithelial cells [62]. In addition, nuclear factor kappa B (NF-κB) represents a family of transcription factors that involve in a wide range of processes similar to MAPKs. Another work also reported urban PM triggered lung inflammation both in vitro and in vivo through the oxidative stress-regulated MAPK/NF-κB pathway [63]. Moreover, a few studies have indicated the cross-talk between the membrane and nuclear receptor signaling pathways with activator protein 1 (AP-1) cascade as a crucial paradigm, direct or indirect, in pulmonary carcinogenesis. There was a report that PM2.5 might induce c-Jun-dependent AP-1 activation through oxidative stress-mediated activation of MAPKs [64]. Ultimately, these signaling pathways together influence the tumor microenvironment, which is important for tumor behavior, as well as the epithelial-mesenchymal transition (EMT), which promotes tumor cell migration and invasion [78,79].

### 3.2. Oxidative Stress-Modulated Transcription Factors

The literature has shown that oxidative stress can activate transcription factors with important roles in lung carcinogenesis. For example, NF-κB is a nuclear factor that induces the expression of crucial genes for lung tumorigenesis, such as anti-apoptosis genes, whereas AP-1 might control a number of cellular processes such as differentiation, proliferation, and apoptosis. Both of these factors were found to be activated by exposure to PM2.5 [80,81]. In contrast, *p53*, which is a well-known tumor-suppressor gene, was be strongly suggested to be inhibited by repeated exposure to PM2.5 through the ROS/AKT-related pathway in human bronchial epithelial cells [65].

### 3.3. Oxidative Stress-Impaired DNA

The impairment of DNA, such as oxidative DNA damage, is considered as one of the major mechanisms of carcinogenesis, particularly in the lungs [82]. In fact, living organisms are always exposed to a variety of risk factors, both exogenous and endogenous, that might be harmful to genome stability and integrity, possibly leading to various types of DNA damage. Luckily, a complicated signaling network including DNA repair and cell cycle checkpoint pathways, also referred to as the DNA damage response (DDR), is activated in response to DNA damage in order to function as an “anti-cancer barrier” [83]. If compromised DNA is inappropriately restored by DDR, the genomic mutations and other irreversible damage will occur, subsequently leading to severe consequences such as malignant progression. Since the lungs are directly exposed to higher oxygen concentrations than other tissues in the human body, the balance between oxidants and antioxidants is vitally important to protect this organ, otherwise oxidative stress will occur, leading to a diversity of adverse events such as chronic inflammation or even DNA impairment that have been demonstrated to be associated with an increased risk of human lung cancer [84]. Thus, the mechanism of PM2.5-induced lung carcinogenesis has been proposed to involve DNA damage caused by oxidative stress and inflammation. Accumulating evidence from cell culture experiments, animal models, and even human studies has indicated that PM2.5 might induce oxidative DNA damage through an oxidative stress paradigm caused by the elevated generation of ROS [66,67,68,69,70,71]. In addition, it is believed that the longer the inflammation persists, the greater the risk of cancer. The underlying mechanisms remain controversial. However, it is well-established that inflammatory cells such as mast cells and leukocytes are recruited to the site of injury, leading to a rapid release of ROS so-called ‘respiratory burst’ during the inflammation process. Simultaneously, inflammatory cells also produce different mediators, such as cytokines and chemokines, to further recruit more inflammatory cells to the site of damage, as well as to activate signaling pathways and/or regulate transcription factors which modulate important immediate cellular stress responses. Otherwise speaking, this sustained oxidative microenvironment resulting from chronic inflammation might disturb the normal cellular homeostasis, in turn capably inducing genomic instability and even carcinogenesis over a long period of time [84]. Once again, there has also been substantial evidence of the capacity of PM2.5 to induce chronic inflammation in the lungs, subsequently leading to oxidative stress and other DNA lesions [72,73,74,75]. Following DNA damage, a DDR or DNA repair process is initiated through a series of cellular processes in order to repair such lesions properly and quickly. Unfortunately, several studies have suggested that PM2.5 possibly decreases the DNA repair capacity [66,69]. Furthermore, recent investigations have found that DDR might be affected by crosstalk with a number of signaling pathways, such as PI3K/AKT, NF-kB and, so on [83]. As mentioned above, PM2.5-induced ROS production has been documented to activate these signaling pathways, further implicating its role in the interruption of DDR. Collectively, PM2.5 exposure might compromise DNA by either one of three mechanisms, including the generation of overwhelming ROS, the development of chronic inflammation and inhibition of DDR, or by a combination of two or all paradigms.

## 4. Antioxidant Therapeutics against PM2.5-Induced Lung Cancer

In general, the human body is protected from the deleterious effects of ROS by diverse antioxidant defense systems, including enzymes and non-enzymes. However, these systems might be overwhelmed under greatly excessive or persistently abnormal production of ROS. Since oxidative stress derived from increased ROS is likely to be one of the major mechanisms of PM2.5-induced lung cancer, antioxidant therapeutics might be promising strategies for cancer prevention. Antioxidant therapeutics can be defined as natural or synthesized products that are capable of neutralizing or preventing oxidative stress. A review, which analyzed the epidemiological literature from 1985 to 1993, independently investigated the effects of dietary intake or serum nutrients of three common natural antioxidants, including carotenoids, vitamin C, and vitamin E, on the risk of lung cancer. All three agents were found to be able to protect humans from lung cancer, in which carotenoids showed a stronger relationship, whereas the associations of the remaining vitamins were weaker [85]. Aside from antioxidant nutrition, the investigation of natural or synthesized compounds that possess antioxidant properties can possibly become an attractive field to fight against cancer, especially lung cancer. However, the publications related to this topic remain scarce. It is well-established that tea-derived polyphenols have antioxidant properties, suggesting that they are capable of regulating oxidative stress. An in vitro study demonstrated the depletion of oxidative stress through decreased PM2.5-generated ROS in human alveolar epithelial cells by tea polyphenols. Furthermore, these substances also significantly inhibited the apoptosis induced by PM2.5 [86]. Another kind of polyphenols, curcumin extracted from the turmeric plant, has also been demonstrated to be able to inhibit both initiation and promotion of cancer through the regulation of ROS and transcription factors such as NF-κB [87,88]. While natural compounds might be beneficial in terms of user-friendly properties and/or biosafety, their bioavailability possibly remains the limitations. Therefore, structure-targeted therapies to achieve optimal or desirable activities have been demanded. Decades ago, carbon monoxide was completely considered as a toxic gas to living organisms. In fact, it is an endogenous byproduct of heme degradation in the body, and has been demonstrated to possess antioxidant properties. Thus, carbon monoxide releasing molecules (CORMs), which are designed to deliver controlled amounts of therapeutic carbon monoxide to target cells and tissues, are expected to be a promising therapy for pathological conditions caused by redox disturbances. Recently, the antioxidant effects and mechanisms of PM2.5 were examined in human pulmonary alveolar epithelial cells and in mice. The authors observed that CORM-2 possibly inhibited not only PM2.5-induced ROS generation, both NADPH oxidase- and mitochondria-derived, but also lung inflammation through the ROS-associated pathway [76]. Furthermore, another work suggested that the combination between antioxidant and other molecules that have effects on the regression of the disease pathogenesis might be more beneficial in the treatment of oxidative stress-induced lung diseases than single antioxidant therapy [89], suggesting that antioxidants probably act as effective supplements.

Based on these preliminary findings, antioxidant products are likely to be beneficial in diminishing the risk of lung cancer induced by PM2.5, providing directions for the development of anti-cancer approaches. However, the limited number of studies, and especially the lack of knowledge regarding the inhibition of the downstream pathway of oxidative stress induced by antioxidants, might hinder our understanding of the actual efficacy of these agents in the protection of human from cancer. Therefore, further investigations are required to establish the protective role of antioxidants in resistance to oxidative stress, potentially resulting in a reduction in the risk of cancer associated with ROS-mediated PM2.5.

## 5. Conclusions

Air pollution has been suspected as a cause of lung cancer for a long time. To date, there is strong evidence to establish a positive association between fine particulate matter, one predominant component of air pollution, with lung cancer process. There is growing documentation that this pollutant might be involved in diverse types of extrapulmonary cancer, making it more and more threatening to human health. Oxidative stress or redox homeostasis imbalance is viewed as one of the plausible mechanisms that evidence PM2.5 carcinogenesis properties. Thus, antioxidant therapeutics possibly play promising roles in cancer treatment and prevention strategies. In summary, the existing literature of the associations between exposure to PM2.5 and cancer progression, as well as the underlying mechanisms, are still far from consistency and conclusiveness. More studies in the future are required to clarify this field, providing insights for the fight against cancer, which is currently one of the major burdens of mankind. 

## Figures and Tables

**Figure 1 cancers-12-02505-f001:**
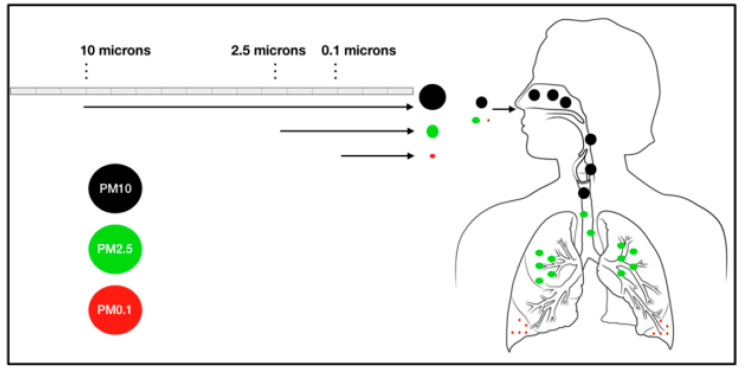
Classification of particulate matter according to aerodynamic diameter. The smaller the particles, the further the penetration and the worse the impact on health. As PM enters into our body upon inhalation, coarse PM or PM10 with a diameter of less than 10 μm is retained in nasal cavities and upper airways, whereas fine PM or PM2.5 with a diameter of less than 2.5 μm and ultrafine PM or PM0.1 with a diameter less than 0.1 μm might travel deeper into lungs and bronchi alveoli, and even invade to further organs or elicit systemic effects through the circulation.

**Figure 2 cancers-12-02505-f002:**
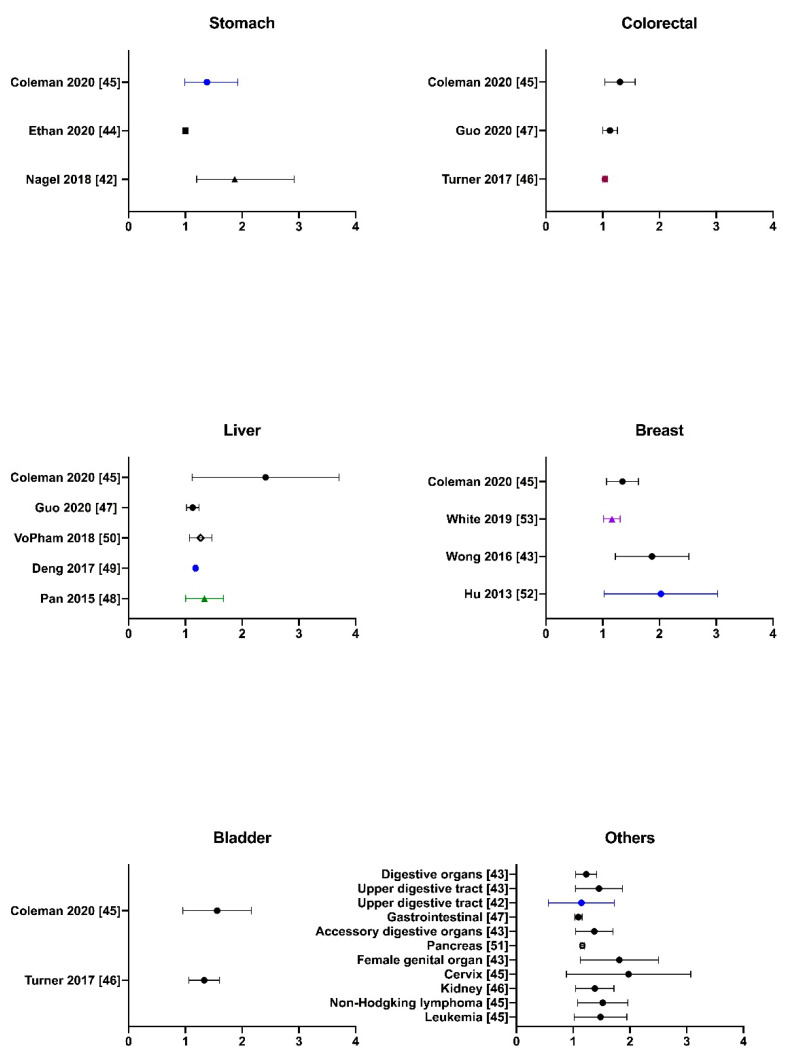
The risk of cancer mortality and/or incidence for extrapulmonary cancer per increment of PM2.5 from 1 μg/m^3^ to 10 μg/m^3^. The illustration is of the comparisons between the hazard ratios (95% CI) of increased PM2.5 with various types of non-lung cancer, including stomach, colorectal, liver, breast, bladder and others. Note: the HR for cancer incidence is symbolized as a triangle, whereas the HR for cancer mortality is symbolized as a circle. There are two exceptional studies that examined the risk ratio, which is marked with a square, or incident rate ratio, which is marked with a diamond. Different colors represent different increments of PM2.5, in which black, blue, pink, purple, and green correspond to 10 μg/m^3^, 5 μg/m^3^, 4.4 μg/m^3^, 3.6 μg/m^3^, and 1 μg/m^3^, respectively.

**Figure 3 cancers-12-02505-f003:**
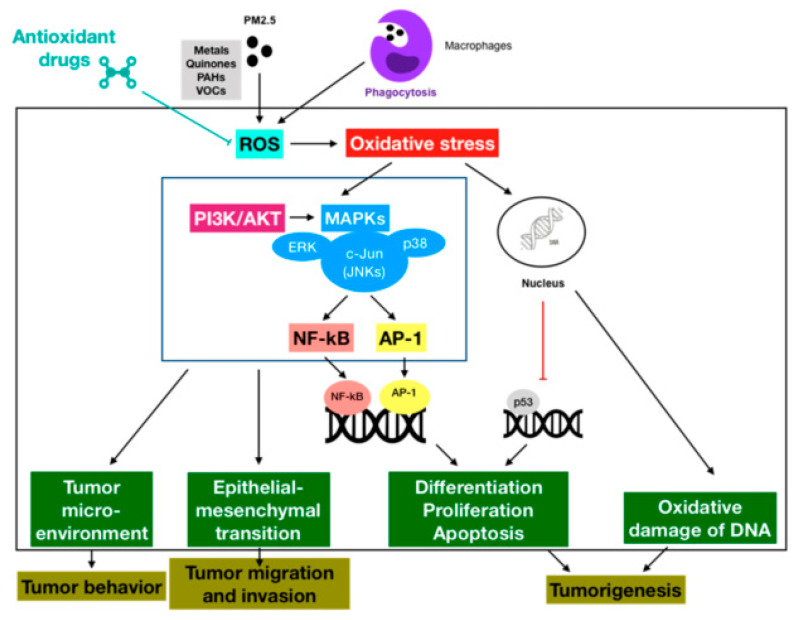
Schematic representation of the cellular pathways of carcinogenic effects induced by oxidative stress through PM2.5-mediated ROS production. Upon chronic exposure to PM2.5, oxidative stress derived from increased ROS production might lead to the activation of signaling pathways, the modulation of transcription factors, and the impairment of DNA, which might together induce tumor initiation and progression. This suggests the potentials of antioxidant therapies as promising strategies in the reduction of cancer diseases caused by redox homeostasis imbalance.

**Table 1 cancers-12-02505-t001:** A summary of experimental studies on the toxicological effects of PM2.5 involved in lung carcinogenesis.

Author, Year	Study Design	Experimental Model	Toxicological Effects of PM2.5	Conclusions
Yang, B. et al. 2016 [35]	In vitro	Human lung carcinoma cell lines A549 and H1299	Enhanced proliferation and motility in both cell lines. Up-regulated IL-1β and MMP-1 genes which were involved in the cell proliferation as well as the progression of invasion and metastasis. Induced mRNA expression of regulated transcripts in response to MAPK signaling pathway.	PM2.5 might induce crosstalk among pathways that promote survival, proliferation, invasion, and migration of cancer cells. MAPK signaling pathway played a crucial role.
Wei, H. et al. 2017 [36]	In vitro	A549 cells	Enhanced cell migration and invasion. Decreased mRNA expression of epithelial markers and increased mRNA expression of mesenchymal markers. Caused changes in the levels of stemness-associated microRNAs. Induced EMT markers and morphology as well as CSC properties.	Chronic PM2.5 exposure possibly induced malignant behaviors as well as EMT and CSC properties in vitro. Stemness-associated microRNAs were likely to be involved.
Xu, H. et al. 2019 [37]	In vitro and In vivo	A549 cells A549 tumor model in female nude mice	Induced secretion of exosomes containing high levels of Wnt3a. Promoted A549 cell proliferation via Wnt3a/β-catenin pathway. Enhanced tumor growth in vivo in Wnt3a-dependent manner.	Exosomes derived from PM2.5 exposure appeared to promote the growth of lung tumor in vitro and in vivo via Wnt3a/β-catenin pathway.
Yang, D. et al. 2017 [38]	In vitro	Human lung carcinoma cell lines A549 and H292	Induced EMT in both cell lines. Up-regulated the activity of Smad1-mediated pathway and down-regulated the expression of Smad6 and Smad7 which were associated with EMT in A549 cells.	PM2.5 might induce EMT through Smad1-mediated signaling pathway.
Yang, J. et al. 2019 [39]	In vivo	Female Kunming mice	Induced lung damage. Increased the number of inflammatory cells, particularly M1 and M2 macrophages. Increased the levels cytokines such as IL-4, TNF-α, TGF-β1 in sera and tissues.	PM2.5 capably activated the inflammatory responses and compromised immune function, leading to ultrastructural damage in mice lungs.
Yang, B. et al. 2018 [40]	In vivo	A549 tumor model in male CB17-SCID mice	Increased the number of tumor nodules. Increased the levels of total protein in BALF. Enhanced the expression of MMP-1, IL-1β and VEGF which are hallmarkers in cancer progression. Increased the levels of angiogenesis factors in blood serum.	PM2.5 possibly promoted lung tumor progression in tumor-bearing mice.
Liu, C.W. et al. 2018 [62]	In vitro In vivo Cross-sectional study	A549 cells WT and IL-6 KO mice COPD patients and Healthy subjects	Induced cytotoxicity, ROS generation and monocyte adherence in A549 cells. Increased the expression of ICAM-1 via IL-6/AKT/STAT3/NF-κB-dependent pathway in A549 cells. Induced the expression of ICAM-1 and IL-6 in lung tissues and plasma of WT mice but not IL-6 KO mice. Higher levels of white blood cell count, CRP, sICAM-1 and IL-6 in COPD patients.	Oxidative stress mediated by PM2.5-induced ROS production might increase the expression of ICAM-1 both in vitro and in vivo. The ICAM-1 expression was regulated through IL-6/AKT/STAT3/NF-κB-dependent pathway in lung epithelial cells.
Wang, J. et al. 2017 [63]	In vitro and In vivo	HBECs Male C57 mice model of PM-induced acute lung inflammation	Enhanced ROS generation in vitro and in vivo. Induced inflammatory responses in vitro through increased the expression of IL-1β, IL-6, IL-8, MMP-9 and COX-2. Activated MAPK (ERK, JNK, p38 MAPK) and NF-κB pathways involving in inflammation in vitro. Increased oxidant stress in lung tissues, infiltration of inflammatory cells, the number of total cells and inflammatory cells as well as the concentrations of IL-1β, IL-6, IL-8, MMP-9 in BALF.	PM2.5 capably induced inflammatory responses both in vitro and in vivo via ROS/MAPK/NF-κB signaling pathway.
Pourazar, J. et al. 2005 [64]	Prospective study	Nonatopic nonsmokers exposed to diesel exhaust	Increased total immunoreactivity of phosphorylated p38 MAPK. Increased nuclear translocation of p65 subunit of NF-κB, c-Jun subunit of AP-1, phosphorylated JNK, and phosphorylated p38 MAPK.	Exposure of healthy subjects to diesel exhaust, a major component of PM2.5, might activate the redox-sensitive transcription factors (NF-κB and AP-1) and related signaling pathways (JNK and p38 MAPK) in bronchial epithelium.
Zhou, W. et al. 2016 [65]	In vitro	HBECs (BEAS-2B)	Induced *p53* promoter hypermethylation and down-regulated p53 expression via DNMT3B up-regulation. Activated ROS-mediated PI3K/AKT pathway which involved in DNMT3B up-regulation.	PM2.5 could induce epigenetic silencing of *p53* through ROS/AKT/DNMT3B pathway-mediated promoter hypermethylation in BEAS-2B cells.
Niu, B.Y. et al. 2020 [66]	In vitro	HBECs (16HBE)	Induced cytotoxicity. Generated oxidative stress through increased levels of oxidants such as ROS and decreased level of antioxidant GSH. Caused DNA damage including DNA strand breaks, oxidative DNA damage and chromatin damage. Altered the expression of DNA repair genes.	PM2.5 capably exerted cytotoxicity and genotoxicity in 16HBE cells through the combined effects of generation of oxidative stress-induced DNA damage and alteration of DDR.
Veerappan, I. et al. 2019 [67]	In vitro	A549 cells	Induced lipid peroxidation and oxidative stress via ROS generation. Induced cytotoxicity and DNA damage. Altered the expression of miRNAs related to oxidative stress, inflammation and DNA damage.	PM2.5 possibly induced genotoxicity in A549 cells by increased production of ROS generation and the alteration of miRNA expression associated to oxidative stress, inflammation and DNA damage.
Ren, X. et al. 2018 [68]	In vitro	HBECs	Decreased cell viability. Induced oxidative DNA double-strand breaks.	PM2.5 might cause cytotoxicity and genotoxicity in HBECs by oxidative manner.
Li, R. et al. 2017 [69]	In vivo	Male Wistar rats	Caused pathological damage in rat lungs. Increased the levels of DNA damage markers. Regulated the expression of DNA repair genes (↑OGG1, ↓MTH1, ↓XRCC1). Altered the levels of oxidative stress markers (↑GADD153, ↑HO-1, ↑MDA, ↓SOD). Increased the levels of metabolic enzymes (CYP450, CYP1A1, CYP1A2, GST).	PM2.5 and higher dosage 9-NA (a typical compound of NPAHs in PM2.5) possibly induced genotoxicity in rat lungs through three major ways including induction of DNA damage combined with inhibition of DNA repair process, production of oxidative stress, and disturbance of biotransformation via metabolic enzymes activation.
Lai, C.H. et al. 2017 [70]	Longitudinal study	Healthy students	Significant relationship between biomarker of urinary exposure (1-OHP) and biomarkers of urinary oxidative and methylated DNA damage (8-oxoG and N7-MeG). N7-MeG increased by 8.1% per 10 μg/m^3^ increment in PM2.5.	PM2.5 might cause oxidative and methylated DNA damage in healthy and young adults.
Tan, C. et al. 2017 [71]	Cross-sectional study	Traffic policemen and Office policemen	Significant correlation between cumulative intersection duty time with biomarkers of oxidative stress (↓GSH) and DNA damage (↑8-OHdG, ↑tail DNA, ↑MN frequency) in traffic policemen. The levels of biomarkers of oxidative stress in traffic policemen were lower than those in office policemen, but of DNA damage were higher.	Long-term exposure to high concentrations of PM2.5 could induce cumulative DNA damage via oxidative stress pathway.
Yuan, Q. et al. 2019 [72]	In vitro	HBECs (HBE and BEAS-2B)	Reduced cell viability. Increased the level of cellular ROS. Induced cellular pro-inflammation through increased levels of IL-6, IL-8. Up-regulated the expression of CYP1A1 and CYP1B1 genes which involved in generation of oxidative stress and pro-inflammation. Inhibition of STAT3/P-STAT3 hampered ROS production and IL6/IL-8 secretion by PM2.5.	PM2.5 capably induced oxidative stress and pro-inflammatory response via up-regulating the expression of CYP1A1/1B1 in two main cell lines of HBECs. JAK/STAT3 pathway might be associated.
He, M. et al. 2017 [73]	In vitro and In vivo	RAW264.7 cells BMDMs of WT and MyD88 deficient BALB/c mice Mouse alveolar cell line MLE-12 BALB/c mice	Induced NF-κB, p38 MAPK and ERK phosphorylation, increased proinflammatory gene and protein expressions, and induced oxidative stress marker HO-1 gene in RAW264.7 cells. Caused increase of proinflammatory mediators in BMDMs in which WT cells were higher than MyD88 cells. Increased proinflammatory gene expressions and HO-1 gene as well as induced intracellular ROS generation in MLE-12 cells. Caused sever alveolitis and bronchitis, and increased the number of macrophages and neutrophils as well as the levels of proinflammatory mediators in mice BALF.	Macrophages might preferentially release proinflammatory mediators due to LPS presenting in PM2.5 via LPS/MyD88 pathway, whereas type II alveolar cell could be prone to PM2.5-induced oxidative stress to cause inflammatory response. PM2.5 possibly induced oxidative stress-dependent inflammation, leading to lung injury.
Riva, D.R. et al. 2019 [74]	In vivo	BALB/c mice	Impaired lung function characterized by increased elastic and viscoelastic components of lung mechanics. Induced lung inflammation via increase in MPO activity, neutrophil infiltration and proinflammatory cytokine expressions (IL-6, TNF-α). Caused oxidative damage expressed by increased reactive substances to thiobarbituric acid and 8-isoprostane. Induced oxidative stress through increase in CAT and reduction in GSH/GSSG.	Acute exposure to low dose of PM2.5 might induce oxidative stress, inflammation and functional impairment in healthy mice lungs.
Shi, Y. et al. 2019 [75]	In vitro and In vivo	HBECs (BEAS-2B) Male Sprague-Dawley rats	Induced genome wide DNA methylation and RNA transcription alterations in BEAS-2B cells. Caused pathological changes such as inflammatory cell infiltration and alveolar wall thickening in rat lung tissue. Enhanced the secretion of cytotoxicity markers and inflammatory cytokines in BALF. Up-regulated the expression levels of inflammatory cytokine genes and NF-κB in BEAS-2B cells and rat lung tissue.	PM2.5 capably induced genome wide DNA methylation and RNA transcription changes as well as inflammatory responses, contributing to pulmonary toxicity and pathogenesis.
Lee, C.W. et al. 2019 [76]	In vitro and In vivo	HPAEpiCs Male BALB/c mice	Induced CRP expression, NLRP3 inflammasome activation, IL-1β secretion, and caspse-1 activation in HPAEpiCs. Induced inflammatory responses via TLR2 and TLR4 in HPAEpiCs. Induced intracellular and mitochondrial ROS generation in HPAEpiCs. Increased the expression of CRP, NLRP3 and ASC protein lung tissue, the levels of IL-1β in serum, and leukocyte count in BALF in mice.	PM2.5 possibly induced lung inflammation via TLR2 and 4/ ROS/NLRPH3 signaling pathway.

Abbreviations: 1-OHP: 1-hydroxypyrene; 8-OHdG: 8-hydroxy-2′-deoxyguanosine; 8-oxoG: 8-oxoguanine; AKT: protein kinase B; AP-1: activator protein 1; ASC: apoptosis-associated speck-like protein; BALF: bronchoalveolar lavage fluid; BMDM: bone marrow-derived macrophage; CAT: catalase; COPD: chronic obstructive pulmonary disease; COX-2: cyclooxygenase 2; CRP: C-reactive protein; CSC: cancer stem cell; CYP: cytochrome P; DDR: DNA damage response; DNMT3B: DNA (cytosine-5-)-methyltransferase 3 beta; EMT: epithelial mesenchymal transition; ERK: extracellular-signal-regulated kinase; GADD153: growth arrest and DNA-damage-inducible gene 153; GSH: glutathione; GST: glutathione S-transferase; HBEC: human bronchial epithelial cell; HO-1: heme oxygenase 1; HPAEpiC: human pulmonary alveolar epithelial cell; ICAM-1: intercellular adhesion molecule 1; IL: interleukin; JAK: janus kinase; JNK: c-Jun N-terminal kinase; KO: knock-out; LPS: lipopolysaccharide; MAPK: mitogen-activated protein kinase; MDA: malondialdehyde; MMP: matrix metalloproteinase; MN: micronuclei; MTH1: MutT homolog 1; N7-MeG: N7-methylguanine; NA: nitroanthracene; NF- κB: nuclear factor kappa B; NLRP3: NOD-, LRR- and pyrin domain-containing protein 3; NPAH: nitro-polycyclic aromatic hydrocarbon; OGG1: 8-oxoguanine DNA glycosylase 1; PI3K: phosphoinositide 3-kinase; ROS: reactive oxygen species; Smad: mothers against decapentaplegic; SOD: superoxide dismutase; STAT3: Signal transducer and activator of transcription 3; TGF-β1: transforming growth factor beta 1; TLR: toll-like receptor: TNF-α: tumor necrosis factor alpha; VEGF: vascular endothelial growth factor; Wnt: wingless-related integration site; WT: wild-type; XRCC1: X-ray repair cross-complementing protein 1.

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
