# Peer review of "The Inducible Role of Ambient Particulate Matter in Cancer Progression via Oxidative Stress-Mediated Reactive Oxygen Species Pathways: A Recent Perception"

_cancers, 2020, doi:10.3390/cancers12092505_

Round 1

Reviewer 1 Report

The topic is interesting, it is one of the main issue in this moment. The problem is that the authors dealt it in no deep way. They did not carefully read the literature, some important contributions about the oxidative stress are missing, especially those related to the PM, ultrafine particles and some elements (e.g., copper) present in such pollutants. Further, they considered the PM0.1 but they should be better if they speak about ultrafine particles with are fundamental aerosol in this filed. The authors should focus their attention on the ultrafine particles which are more dangerous than the other size fractions. The role of PM10 on cancer is well-known as well as the PM2.5 whereas the ultrafine particles are the new target in this field. I suggest to re-model the paper highlighting the role and importance of these fractions.

Author Response

We appreciate your precious comments. We agree that PM0.1 seems to be more harmful than larger PM fractions. However, the data regarding to its health effects on human remain scarce. Meanwhile, PM2.5 is currently chosen to provide international recommendation regarding to health effects because it might best describe the components of PM responsible for adverse conditions. Moreover, the negative effects of PM2.5 can partly overlap those of PM0.1 since PM2.5 also consists of smaller fraction PM0.1. To our knowledge, there is no specific review about the impact of PM2.5 as a single risk factor on cancer which is a major health burden in mankind. Thus, we tried to summarize and analyze the most recent findings in this area, particularly the associations between PM2.5 and various types of cancer along with the oxidative stress mechanisms as its possible carcinogenic mechanisms. We introduced the potential of PM0.1 to cause adverse health effects and mentioned the demand for further studies concerning this field. The reasons why we specifically focused on the health effects of PM2.5 were presented. In the future, we hope that we would perform another review on the role and the importance of the finest fraction of PM on the pathogenesis in human.

Reviewer 2 Report

The review of scientific publications is very comprehensive and thorough. Undoubtedly, this paper is very valuable. It summarizes data from a large number of publications on the effects of particulate matter on cancer.

Minor comments:

Line 35-36. Sentence “Unfortunately, everyone breaths thousands of liters of air daily” is little bit misunderstanding.

Line 35-36. Sentence “Otherwise speaking, our body uptakes and accumulates significant doses of these carcinogens over time” in general not correct.

Line 45. “Generally, PM is inhaled through nasal airway”. In general, this statement is correct. However, did not mentioned what is the difference between breathing through the nose and mouth? Is the amount of particulate matter inhaled different?

Line 204-206. “Moreover, finer particles or aerosols, 205 which carry toxic substances, might be absorbed into bloodstream or delivered to extrapulmonary organs through blood circulation”. This statement is correct in principle. However, in the context of his paper, and in particular section 2.3, it may be inaccurate.

Was a clear distinction between PM2.5 and PM0.1 in the publications discussed in this article? Since ultrafine par­ticles (PM0.1) can pass through the alveoli and spread to other organs, whereas PM2.5 deposits in alveoli. I suggest discussing this issue briefly in this section.

Author Response

  1. Line 35-36. Sentence “Unfortunately, everyone breaths thousands of liters of air daily” is little bit misunderstanding.

Responses to Comments:

Thank you very much for Reviewer’s critical comments. We have corrected this sentence into “In fact, everyone breaths thousands of liters of air daily” to avoid misunderstanding.

  1. Line 35-36. Sentence “Otherwise speaking, our body uptakes and accumulates significant doses of these carcinogens over time” in general not correct.

Responses to Comments:

Thank you very much for Reviewer’s critical comments. We have corrected this sentence into “This implies that humans might be at greater risk of developing cancer if continuously expose to significant concentrations of polluted air, particularly particulate matter, over time” for more accuracy.

  1. Line 45. “Generally, PM is inhaled through nasal airway”. In general, this statement is correct. However, did not mentioned what is the difference between breathing through the nose and mouth? Is the amount of particulate matter inhaled different?

Responses to Comments:

Thank you very much for Reviewer’s critical comments. We have corrected this sentence into “Generally, PM is introduced into human body through inhalation process” to avoid confusion. We aim to describe the gain access of PM into body rather than to emphasize the difference between nose breathing and mouth breathing.

  1. Line 204-206. “Moreover, finer particles or aerosols, 205 which carry toxic substances, might be absorbed into bloodstream or delivered to extrapulmonary organs through blood circulation”. This statement is correct in principle. However, in the context of his paper, and in particular section 2.3, it may be inaccurate.

Responses to Comments:

Thank you very much for Reviewer’s critical comments. In our opinion, section 2.3 is about literature evidence of PM2.5-associated non-lung cancer. Thus, this statement is pretty important to indicate the potentials of PM2.5 to further travel into other organs in additional to lungs, subsequently eliciting harmful effects on extrapulmonary parts. We also provided experimental evidence on the presence of PM2.5 in mice kidney and liver to support this statement. We did no correction in response to this comment.

  1. Was a clear distinction between PM2.5 and PM0.1 in the publications discussed in this article? Since ultrafine par­ticles (PM0.1) can pass through the alveoli and spread to other organs, whereas PM2.5 deposits in alveoli. I suggest discussing this issue briefly in this section.

Responses to Comments:

Thank you very much for Reviewer’s critical comments. The clear distinction between PM2.5 and PM0.1 yet to be elucidated since there are overlap among different particles. We have introduced more about PM0.1 and mentioned the demand for further investigations regarding to this finest fraction.

Round 2

Reviewer 1 Report

The authors gave me their evaluation about the choice of PM2.2. I am still convinced that PM2.5 is not the best index for describing a relationship with the cancer but they gave me satisfying reasons.I think that the paper could be accepted in this form.